# Sentinel Lymph Node Biopsy in Breast Cancer Patients Undergoing Neo-Adjuvant Chemotherapy: Clinical Experience with Node-Negative and Node-Positive Disease Prior to Systemic Therapy

**DOI:** 10.3390/cancers15061719

**Published:** 2023-03-11

**Authors:** Corrado Tinterri, Andrea Sagona, Erika Barbieri, Simone Di Maria Grimaldi, Giulia Caraceni, Giacomo Ambrogi, Flavia Jacobs, Ersilia Biondi, Lorenzo Scardina, Damiano Gentile

**Affiliations:** 1Breast Unit, IRCCS Humanitas Research Hospital, Via Manzoni 56, 20089 Milan, Italy; 2Department of Biomedical Sciences, Humanitas University, Via Rita Levi Montalcini 4, 20090 Milan, Italy; 3Medical Oncology and Hematology Unit, IRCCS Humanitas Research Hospital, Via Manzoni 56, 20089 Milan, Italy; 4Division of Breast Surgery, Department of Woman and Child Health and Public Health, IRCCS A. Gemelli University Polyclinic Foundation, Sacred Heart Catholic University, 00168 Rome, Italy

**Keywords:** sentinel lymph node biopsy, breast cancer, neo-adjuvant chemotherapy, surgery

## Abstract

**Simple Summary:**

Axillary status is crucial for determining the correct local and systemic treatment. The possibility of de-escalating axillary surgery in patients with breast cancer undergoing neo-adjuvant chemotherapy is controversial. This is especially true for clinically node-positive (cN+) patients, for whom axillary lymph node dissection still represents the gold standard, in contrast to clinically node-negative (cN0) patients, for whom sentinel lymph node biopsy has become more widely accepted. Several studies have recently shown that a minimally invasive surgical approach of the axilla is safe in cN+ patients who become cN0 after neo-adjuvant chemotherapy, raising new questions about the potential benefit of this strategy. This retrospective study is aimed at assessing the reliability of this approach by comparing the characteristics and oncological outcomes (e.g., overall survival) of cN0 and cN+ patients before neo-adjuvant chemotherapy and axillary surgery type.

**Abstract:**

Background: Sentinel lymph node biopsy (SLNB) has emerged as the standard procedure to replace axillary lymph node dissection (ALND) in breast cancer (BC) patients undergoing neo-adjuvant chemotherapy (NAC). SLNB is accepted in clinically node-negative (cN0) patients; however, its role in clinically node-positive (cN+) patients is debatable. Methods: We performed a retrospective analysis of BC patients undergoing NAC and SLNB. Our aim was to evaluate the clinical significance of SLNB in the setting of NAC. This was accomplished by comparing the characteristics and oncological outcomes between cN0 and cN+ patients prior to NAC and type of axillary surgery. Results: A total of 291 patients were included in the analysis: 131 were cN0 and 160 were cN+ who became ycN0 after NAC. At a median follow-up of 43 months, axillary recurrence occurred in three cN0 (2.3%) and two cN+ (1.3%) patients. However, there were no statistically significant differences in oncological outcomes (disease-free survival, distant disease-free survival, overall survival, and breast-cancer-specific survival) between cN0 and cN+ patients nor between patients treated with SLNB only or ALND. Conclusions: SLNB in the setting of NAC is an acceptable procedure with a general good prognosis and low axillary failure rates for both cN0 and cN+ patients.

## 1. Introduction

Lymphatic mapping with sentinel lymph node biopsy (SLNB) has emerged as the standard technique to replace routine axillary lymph node dissection (ALND) in early breast cancer (BC) surgery, especially for clinically node-negative (cN0) axilla cases [1,2,3]. Eventually, ALND omission has also been adopted for patients with T1-2, cN0 early BC with one to two positive sentinel lymph nodes (SLN) undergoing breast-conserving surgery [4,5]. The SLNB technique is a minimally invasive approach and provides the same staging information that is gathered with ALND, but with a negligible arm morbidity [1]. In patients with advanced loco-regional BC, the role of neo-adjuvant chemotherapy (NAC) as a means of down-staging the axilla has been investigated both to minimize the extent of axillary surgery and to avoid the complications of ALND [6,7,8,9,10]. However, the different role and oncological consequences of performing SLNB in patients with either cN0 or clinically node-positive (cN+) disease prior to NAC are still unclear. It remains debatable whether SLNB should be performed in patients with BC who are candidates for NAC, as pre-operative systemic treatment could potentially alter the lymphatic drainage from the breast [9,11]. At present, SLNB is used for axillary staging in patients with cN0 disease prior to NAC, with studies showing false-negative rates (FNRs) of approximately 10% and acceptable SLN identification rates [12,13,14,15,16]. The use of anthracyclines and taxane-based chemotherapy regimens, in combination with Trastuzumab and Pertuzumab in the case of human epidermal growth factor receptor 2 (HER2)-positive disease, has shown that depending on the biological BC subtype, nodal involvement can be eradicated in approximately 50–75% of patients [7,13,17,18,19]. However, the use of SLNB following NAC for patients who initially had cN+ disease and then converted to cN0 has been questioned, as previous studies reported FNRs ranging from 8.4% to 23.9% [8,9,10,20,21,22]. Target axillary dissection (TAD) has been suggested to reduce the FNR by marking before NAC the positive axillary nodes with a tattoo/metallic clip, using a dual tracer, or removing at least three SLNs [23,24,25]. However, none of the studies describing these techniques used to reduce the FNR have examined the prognostic significance on patients. On the other hand, numerous studies showed that the use of SLNB only was not associated with an increase in axillary failures or worsening oncological results [3,26,27,28,29]. Based on this background, the aim of this retrospective study was to assess the reliability and clinical significance of lymphatic mapping with SLNB in the setting of NAC comparing the characteristics and oncological outcomes between cN0 and cN+ patients prior to chemotherapy and type of axillary surgery.

## 2. Materials and Methods

### 2.1. Study Design and Patient Management

A retrospective review of all the consecutive patients with BC scheduled for NAC and SLNB at the Breast Unit of IRCCS Humanitas Research Hospital (Milan, Italy) between November 2008 and December 2021 was conducted. Each patient gave the informed consent for surgery and clinical data collection. Pre-operative tumor extension was determined in all patients by clinical examination and bilateral breast and axillary ultrasound (US). Mammography and magnetic resonance imaging (MRI) of the breast were not mandatory but were performed in most cases. In cases of suspicious axillary lymph nodes, either one or a combination of fine needle aspiration, core needle biopsy, or positron emission tomography (PET) scan was performed to assess the presence of metastases. A multidisciplinary tumor board composed of breast surgeons, breast oncologists, breast pathologists, radiotherapists, radiologists, and plastic surgeons discussed the management of every BC patient scheduled for NAC and SLNB. Pre-operative systemic therapy regimens were determined according to the BC biological subtype and hospital protocols, which are based on international guidelines. Clinical and radiological re-staging after NAC was mandatory in all patients. Physical examination was performed after each cycle of chemotherapy. After three months of NAC, all patients underwent bilateral breast and axillary US re-evaluation. Patients who had previously undergone PET were re-examined with the same imaging technique. All patients remained or became ycN0 at the end of NAC and underwent surgical treatment. Surgery was performed within 30 days of the last NAC cycle. The following exclusion criteria were used: patients with BC scheduled for NAC and direct ALND, disease progression during NAC (defined by RECIST criteria [30]), patients with other prior malignancies, follow-up < 12 months, and lost to follow-up.

### 2.2. Lymphatic Mapping, Sentinel Lymph Node Biopsy, and Axillary Treatment

The protocol of lymphatic mapping was performed with the use of radioisotope alone; TAD was never used. For palpable tumors, the standard technique included a peri-tumoral injection of 99Technetium labeled radiocolloid, in an effort to replicate the intra-mammary lymphatic pathways potentially traversed by metastases. On the other hand, for non-palpable or multicentric tumors, a dermal injection of 99Technetium-labeled radiocolloid in the sub-areolar plexus was performed. A pre-operative lymphoscintigraphy was performed the day before or the same day of surgery. To reduce the length of the surgical incision and to minimize the extent of the dissection, the location of the SLN was marked on the intact skin by indelible ink. Before incision, the location of the radioactive SLN was further determined with the use of a gamma probe through the intact skin. The site and direction of the surgical incision were chosen based on this knowledge. Usually, an incision of a few centimeters was sufficient for SLN identification. Intra-operatively, the gamma probe was inserted into the axillary wound to guide the direction of the dissection until the SLN was identified. All patients who remained or became ycN0 after NAC underwent SLNB; at least one SLN was found in all cases. Once the SLN was identified using the gamma probe, it was removed and sent for complete intra-operative frozen section pathological examination. In all ypN0 patients, ALND was omitted. Complete ALND was performed only if the SLN was positive at intra-operative pathological evaluation. From December 2018, patients began enrollment in the NEONOD2 prospective clinical trial, and in case of micrometastatic SLN and ALND were omitted [31].

### 2.3. Statistical Analysis

Patients were selected from the institutional database (the last follow-up was current up to 9 December 2022). The patient characteristics are presented according to initial axillary status (cN0 versus cN+) and reported as median and range for continuous variables and frequencies (no., %) for categorical variables. Differences in demographic, clinic-pathologic, and treatment characteristics between the two groups (cN0 versus cN+) were compared by using the chi-square test or Fisher’s exact test. For recurrence and survival analyses, patients were divided into different groups based on their axillary status prior to NAC (cN0 versus cN+) and type of axillary surgery (SLNB only versus ALND). Disease-free survival (DFS) was defined as the period from the date of surgical treatment to the date of any tumor progression including loco-regional recurrence or distant metastasis. Distant disease-free survival (DDFS) was defined as the period from the date of surgery to the date of detection of distant metastasis. Overall survival (OS) was defined as the time interval from surgical treatment to death from any cause or to the date of last contact. Finally, breast-cancer-specific survival (BCSS) was determined by selecting BC as the cause of death and recording the follow-up duration after censoring deaths from other causes. The Kaplan–Meier method was used to generate the recurrence and survival curves and to estimate the DFS, DDFS, OS, and BCSS rates. Statistical significance was set at *p* < 0.05 and all statistical tests were two-tailed. 

## 3. Results

### 3.1. Characteristics of Patients and Axillary Surgery

A total of 291 cT1-4 and cN0-cN+ patients with BC treated with NAC and SLNB were included in the study: 131 cN0 who remained ycN0 and 160 cN+ who became ycN0 after NAC. The median age was 48 years (range, 28–79) in cN0 patients and 50 years (range, 29–87) in cN+ patients. Overall, 63 cN0 (48.1%) and 81 cN+ (50.6%) patients were post-menopausal at the time of surgery. The median diameter of the breast tumor prior to NAC was 30 mm in both groups (*p* = 0.288). cT2 tumors (cN0 67.9% versus cN+ 68.8%, *p* = 0.627) were presented for the majority of patients. There were no statistically significant differences in terms of patient and tumor characteristics, NAC regimens, and post-operative treatments between cN0 and cN+ patients. However, cN+ patients were more frequently treated with ALND compared with cN0 patients (*p* = 0.001). Demographic, clinic-pathologic characteristics, and treatments received according to axillary status prior to NAC are shown in Table 1.

A median of 1 SLN was removed in each group. No patient with an intra-operative negative SLN became positive at final histopathology. Overall, cN+ patients were more likely to still have SLN involvement after NAC (cN0 14.5% versus 36.3% cN+, *p* < 0.001); however, with the use of NAC, ALND was avoided in 113 cN+ (70.6%) patients. A nodal pathologic complete response defined as ypN0 was achieved in 102 cN+ (63.8%) patients. Of 13 cN+ patients who became ypNi+/mi after NAC, 9 were enrolled in the NEONOD2 trial [31] and treated with SLNB only. Among patients treated with ALND, the median number of axillary lymph nodes examined was 12 (range, 3–28) in cN0 patients and 13 (range, 5–27) in cN+ patients. Of note, 12 cN+ (7.5%) patients treated with ALND had more than three additional positive lymph nodes at definitive pathological evaluation. Axillary lymph node characteristics of patients are detailed in Table 2.

### 3.2. Oncological Outcomes

At a median follow-up of 43 months (range, 12–169), axillary recurrence occurred in three cN0 (2.3%) and two cN+ (1.3%) patients. Of the three cN0 axillary recurrences, one was synchronous with an ipsilateral breast recurrence that occurred 10 months after the initial surgery and was treated with mastectomy and ALND, while the other two patients experienced an isolated axillary recurrence that was treated with ALND. Of the two cN+ axillary recurrences, one was synchronous with a post-mastectomy surgical scar recurrence and was treated with ALND and skin excision 15 months after the first surgery, while the other patient had an isolated axillary recurrence and was treated with ALND 58 months after the first surgery. At the last follow-up visit, these five patients who had developed axillary recurrence were alive without evidence of disease. Overall, 22/291 (7.6%) patients died, of which 14/131 were cN0 (10.7%) and 8/160 were cN+ (5.0%). Patients were divided according to their axillary status prior to NAC (cN0 versus cN+) and type of axillary surgery performed (SLNB only versus ALND), and their oncological results were compared. The DFS rates at 1, 3, and 5 years were 94.6%, 85.6%, and 82.8% and 97.5%, 91.3%, and 83.6% in cN0 and cN+ patients, respectively. The DDFS rates at 1, 3, and 5 years were 96.9%, 88.5%, and 80.6% and 98.8%, 92.2%, and 87.5% in cN0 and cN+ patients, respectively. The OS rates at 1, 3, and 5 years were 99.2%, 94.2%, and 89.6% and 98.1%, 96.4%, and 93.2% in cN0 and cN+ patients, respectively. The BCSS rates at 1, 3, and 5 years were 100%, 94.9%, and 93.3% and 99.4%, 98.3%, and 96.7% in cN0 and cN+ patients, respectively. The DFS rates at 1, 3, and 5 years were 96.0%, 88.4%, and 84.0% and 96.9%, 90.0%, and 82.8% in patients treated with SLNB only or ALND, respectively. The DDFS rates at 1, 3, and 5 years were 97.8%, 90.9%, and 83.5% and 96.9%, 89.3%, and 86.3% in patients treated with SLNB only or ALND, respectively. The OS rates at 1, 3, and 5 years were 98.7%, 95.0%, and 91.7% and 98.5%, 96.8%, and 90.9% in patients treated with SLNB only or ALND, respectively. The BCSS rates at 1, 3, and 5 years were 99.6%, 96.4%, and 94.1% and 100%, 98.3%, and 98.3% in patients treated with SLNB only or ALND, respectively. There were no statistically significant differences in terms of DFS, DDFS, OS, and BCSS between cN0 and cN+ patients prior to NAC (*p* = 0.180, *p* = 0.280, *p* = 0.181, and *p* = 0.102, respectively), nor between patients treated with SLNB only or ALND (*p* = 0.669, *p* = 0.665, *p* = 0.429, and *p* = 0.776, respectively). Figure 1 and Figure 2 show the Kaplan–Meier recurrence and survival curves.

## 4. Discussion

The findings of this retrospective study contribute to the existing literature showing that lymphatic mapping with SLNB in the setting of NAC presents a good prognosis with low rates of axillary relapses in patients with cN0 or cN+ BC. Axillary recurrence rates are consistent with those of previous retrospective analyses, ranging from 0% to 2.3% [26,27,28,29].

It is generally accepted that SLNB after NAC is performed in cN0 patients at the time of presentation, considering that FNRs are similar to patients with BC undergoing upfront surgery [12,13,14,15,16]. However, several studies have reported high FNRs (>10%) for SLNB in cN+ patients who converted to cN0 after NAC [8,9,21,22]. The American College of Surgeons Oncology Group Z1071 prospective clinical trial [8] enrolled 756 cN+ patients receiving NAC from 136 institutions. Eventually, 649 of them underwent pre-operative systemic therapy followed by both SLNB and ALND, resulting in an FNR of 12.6%. The same issue was explored in the four-arm, prospective, multicenter SENTINA trial [9], which enrolled 1737 patients from 103 institutions. Patients with cN+ disease received NAC, and those who converted to ycN0 (arm C) were treated with SLNB and ALND. The FNR was 24.3% for patients who had one SLN removed and 18.5% for those who had two SLNs removed. In 2015, van Nijnatten et al. [21] published a systematic review and meta-analysis investigating the use of SLNB after NAC in patients with cN+ BC. They included eight studies in the analysis; the pooled estimate of FNR was 15.1%. After subgroup analysis, FNR was significantly worse if one SLN was removed compared with two or more SLNs (23.9% versus 10.4%, *p* = 0.026). In 2019, Simons et al. [22] analyzed 20 unique studies with 2217 included patients; the FNR was 17%. For this reason, lymphatic mapping with SLNB has conventionally not been recommended as an alternative axillary staging procedure to routine ALND. 

One of the strategies suggested by the St. Gallen 2021 Consensus [32] to reduce the FNR in cN+ patients converted to cN0 after NAC was to remove more than three SLNs. Similarly, the 2022 NCCN guidelines [33] proposed TAD as an alternative to reduce FNR, marking the positive axillary lymph nodes with a clip or using a dual tracer. However, the evidence supporting the need for localization and retrieval of the clipped lymph nodes is less robust and mostly depends on retrospective analyses, particularly in patients who had suboptimal lymphatic mapping procedures [23,34]. Moreover, failure to retrieve the clipped lymph nodes has been reported in up to 30% of cases [35,36], raising the question of whether complete ALND is therefore necessary.

We believe that none of the efforts to lower the FNR are clinically significant in terms of prognosis; in fact, none of the aforementioned studies examined the oncological results of the patients. Here, on the contrary, the oncological outcomes were reviewed, and, in light of the results, we can state that lymphatic mapping with SLNB can be performed in cN0 or cN+ patients who are eligible for NAC. This concept has already been described by Kahler-Riberiro-Fontana et al. [26], who analyzed consecutive cT1-3, cN0, or cN+ patients operated on at the European Institute of Oncology who became or remained cN0 after NAC and underwent SLNB. In fact, after a median follow-up of 9.2 years, axillary failure occurred only in 1.8% of the initially cN+ patients and in 1.5% of the initially cN0 patients, with OS rates at 5 and 10 years in the whole cohort of 91.3% and 81.0%, respectively. Additionally, in a recent retrospective study, Barrio et al. [27] analyzed 610 consecutively identified cT1-3, cN+ patients who converted to cN0 after NAC and underwent SLNB with dual tracer mapping. At a median follow-up of 40 months, there was one nodal recurrence synchronous with a local recurrence in a patient who refused radiotherapy. The DDFS and OS rates at 5 years were 92.7% and 94.2%, respectively.

Our study has some limitations. First, it is a single-institution study, which has limitations due to its retrospective design and observational data collected at a specific time point. Second, the study had a short median follow-up. Additionally, no power analysis was performed to determine the optimal sample size to detect statistical significance.

## 5. Conclusions

In conclusion, the breast unit is moving towards a de-escalation of axillary surgery even in the setting of NAC. The results demonstrated that lymphatic mapping with SLNB maintained its expected staging and prognostic role. Sentinel lymph node biopsy is an acceptable procedure with a general good prognosis and low axillary failure rates in patients with cN0 or cN+ BC undergoing NAC who subsequently remained or became ycN0. The use of SLNB only in these patients did not translate into impaired oncological outcomes; however, further studies are needed to examine what effect lowering the FNR after SLNB only could have on the prognosis of these patients.

## Figures and Tables

**Figure 1 cancers-15-01719-f001:**
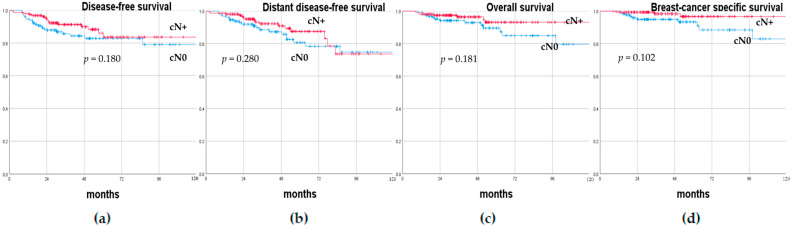
(**a**) Disease-free survival, (**b**) distant disease-free survival, (**c**) overall survival, (**d**) and breast-cancer-specific survival curves of breast cancer patients according to their axillary status prior to neo-adjuvant chemotherapy (cN0 versus cN+).

**Figure 2 cancers-15-01719-f002:**
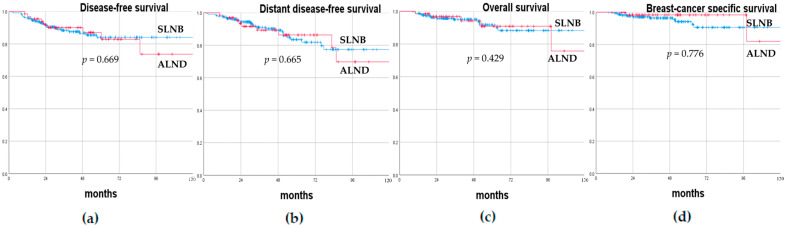
(**a**) Disease-free survival, (**b**) distant disease-free survival, (**c**) overall survival, (**d**) and breast-cancer-specific survival curves of breast cancer patients undergoing neo-adjuvant chemotherapy according to the type of axillary surgery performed (sentinel lymph node biopsy only versus axillary lymph node dissection).

**Table 1 cancers-15-01719-t001:** Demographic, clinic-pathologic characteristics, and treatment data of breast cancer patients according to cN status prior to neo-adjuvant chemotherapy.

	**cN0 (No. 131)** **Tot. (%)/** **Median (Range)**	**cN+ (No. 160)** **Tot. (%)/** **Median (Range)**	**Univariate Analysis** ***p*-Value**
Patient			
Age (years)	48 (28–79)	50 (29–87)	0.144
Post-menopausal	63 (48.1%)	81 (50.6%)	0.668
Tumor			
Dimension pre-NAC (mm)	30 (12–90)	30 (7–100)	0.288
Clinical T pre-NAC			
- cT1b	0 (0%)	1 (0.6%)	0.627
- cT1c	23 (17.6%)	24 (15.0%)	-
- cT2	89 (67.9%)	110 (68.8%)	-
- cT3	15 (11.5%)	19 (11.9%)	-
- cT4	4 (3.0%)	6 (3.7%)	-
- cM1	0 (0%)	2 (1.3%)	0.200
Multifocality/multicentricity	30 (22.9%)	55 (34.4%)	0.469
Anthracycline only	10 (7.6%)	7 (4.4%)	0.094
Anthracycline and taxanes	67 (51.2%)	75 (46.9%)	-
Anthracycline, taxanes, and Trastuzumab	53 (40.5%)	75 (46.9%)	-
Anthracycline, taxanes, and Pertuzumab	1 (0.7%)	2 (1.3%)	-
CDK inhibitor	0 (0%)	1 (0.6%)	-
Subtype			
- Luminal-like	32 (24.4%)	43 (26.9%)	0.715
- HER2+	56 (42.8%)	78 (48.8%)	-
- Triple negative	43 (32.8%)	39 (24.3%)	-
Histotype			
- Ductal	127 (97.0%)	151 (94.4%)	0.366
- Lobular	2 (1.5%)	7 (4.4%)	-
- Mucinous	2 (1.5%)	1 (0.6%)	-
- Papillary	0 (0%)	1 (0.6%)	-
Vascular invasion	22 (16.8%)	25 (15.6%)	0.788
Ki67	15 (2–90)	10 (1–85)	0.091
Dimension post-NAC (mm)	8 (0–70)	7 (0–60)	0.086
Pathological T post-NAC			
- ypT0	22 (16.8%)	38 (23.8%)	0.051
- ypTis	19 (14.5%)	20 (12.5%)	-
- ypTmi	2 (1.5%)	8 (5.0%)	-
- ypT1a	8 (6.1%)	8 (5.0%)	-
- ypT1b	22 (16.8%)	26 (16.3%)	-
- ypT1c	27 (20.6%)	39 (24.3%)	-
- ypT2	28 (21.5%)	20 (12.5%)	-
- ypT3	2 (1.5%)	1 (0.6%)	-
- ypT4	1 (0.7%)	0 (0%)	-
Surgery			
- BCS	81 (61.8%)	100 (62.5%)	0.907
- Mastectomy	50 (38.2%)	60 (37.5%)	-
- ALND	18 (13.7%)	47 (29.4%)	0.001 ^a^
Post-operative treatment			
- Chemotherapy	20 (15.3%)	18 (11.3%)	0.313
- Radiotherapy	97 (74.1%)	133 (83.1%)	0.059
- Endocrine	66 (50.4%)	86 (53.8%)	0.569
- T-DM1	34 (26.0%)	42 (26.3%)	0.434

Footnotes: cN0, clinically node-negative; cN+, clinically node-positive; NAC, neo-adjuvant chemotherapy; CDK, cyclin-dependent kinase; HER2, HER2 evaluated either on immunohistochemistry or on in situ hybridization, according to the ASCO CAP guidelines; BCS, breast-conserving surgery; ALND, axillary lymph node dissection; T-DM1, Trastuzumab emtansine; a, statistically significant.

**Table 2 cancers-15-01719-t002:** Axillary lymph node characteristics of breast cancer patients according to cN status prior to neo-adjuvant chemotherapy.

	**cN0 (No. 131)** **Tot. (%)/** **Median (Range)**	**cN+ (No. 160)** **Tot. (%)/** **Median (Range)**	**Univariate Analysis** ***p*-Value**
Intra-operative SLN status			
Number of SLNs	1 (1–5)	1 (1–6)	0.004 ^a^
Number of patients with positive SLNs	19 (14.5%)	58 (36.3%)	<0.001 ^a^
Pathological N post-NAC			
- ypN0	112 (85.5%)	102 (63.8%)	<0.001 ^a^
- ypNi+	0 (0%)	1 (0.6%)	-
- ypNmi	5 (3.8%)	12 (7.5%)	-
- ypN1a	11 (8.4%)	29 (18.1%)	-
- ypN2a	1 (0.8%)	13 (8.1%)	-
- ypN3a	2 (1.5%)	3 (1.9%)	-
Non-SLN status at pathological evaluation			
Number of evaluated non-SLNs	12 (3–28)	13 (5–27)	0.903
Number of positive non-SLNs	0 (0–13)	1 (0–26)	0.516
Number of patients with 1 positive non-SLN	3 (2.3%)	7 (4.4%)	0.140
Number of patients with 2 positive non-SLNs	1 (0.8%)	6 (3.7%)	-
Number of patients with 3 positive non-SLNs	0 (0%)	3 (1.9%)	-
Number of patients with >3 positive non-SLNs	3 (2.3%)	12 (7.5%)	-

Footnotes: cN0, clinically node-negative; cN+, clinically node-positive; SLN, sentinel lymph node; NAC, neo-adjuvant chemotherapy; a, statistically significant.

## Data Availability

Data supporting reported results can be found in Appendix A.

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
