# Peer review of "Sentinel Lymph Node Biopsy in Breast Cancer Patients Undergoing Neo-Adjuvant Chemotherapy: Clinical Experience with Node-Negative and Node-Positive Disease Prior to Systemic Therapy"

_cancers, 2023, doi:10.3390/cancers15061719_

Round 1

Reviewer 1 Report

Thank you for letting me review this manuscript. I have some major concerns, though.

It is difficult to follow the analyses. E.g., In the Methods the authors state that all patients that remained or became ycN0 underwent SLNB and ALND was omitted in those who were ypN0. ALND was performed only if the SLN was positive at intraoperative examination. Was there no patient with an intraoperative negative SLN that became positive at final histpopathology?

In table 2, the total number of cN0 adds up to 132 and the number of patients with N+ adds up to 19 instead of 20. One cN+patients was classified as ypNi+ but registred as node negative according to the numbers.

The survival ananlyses were made comparing preoperative clinical node status. The analyses shows that the prognosis is very good in general but it does not say anything about SLNB being equivalent to ALND as different patients groups are included recieving different treatment.

Line 70-76 in the introduction need to be re-written. It does not refere to  data that shows that  lowering the FNR does not have a prognostic significance. Neither does this study.

As this retrospective study did not include any power calculations, it can only be seen as hypothesis generating. The Conclusion is too far stretching concerning the safety of SLNB in this patient group.

It would be of the greatest interest to examine what effect lowering the false negative rate after SLNB only, could have on prognosis in these patients . We now know what the authors believe (Discussion line 257) but scientific evidence would be more interesting.

Author Response

POINT-BY-POINT REPLY TO REVIEWERS’ COMMENTS

We thank the editors and reviewers of CANCERS for the opportunity to reply to reviewers’ comments. We do believe that now the manuscript is much more precise and will be of interest to the readers of CANCERS. Please find our replies in this document in bold black.

Reviewers' Comments & Replies:

Reviewer #1: Thank you for letting me review this manuscript. I have some major concerns, though.

It is difficult to follow the analyses. E.g., In the Methods the authors state that all patients that remained or became ycN0 underwent SLNB and ALND was omitted in those who were ypN0. ALND was performed only if the SLN was positive at intraoperative examination. Was there no patient with an intraoperative negative SLN that became positive at final histpopathology?

Reply: We thank the reviewer for the comment.

No patient with an intra-operative negative SLN became positive at definitive histopathology examination.

This was added for clarification in the results section.

In table 2, the total number of cN0 adds up to 132 and the number of patients with N+ adds up to 19 instead of 20. One cN+patients was classified as ypNi+ but registred as node negative according to the numbers.

Reply: We thank the reviewer for the comment.

Table 2 was corrected.

The survival ananlyses were made comparing preoperative clinical node status. The analyses shows that the prognosis is very good in general but it does not say anything about SLNB being equivalent to ALND as different patients groups are included recieving different treatment.

Reply: We thank the reviewer for the comment.

The discussion and conclusion were modified.

Line 70-76 in the introduction need to be re-written. It does not refere to  data that shows that  lowering the FNR does not have a prognostic significance. Neither does this study.

Reply: We thank the reviewer for the comment.

The introduction was modified.

As this retrospective study did not include any power calculations, it can only be seen as hypothesis generating. The Conclusion is too far stretching concerning the safety of SLNB in this patient group.

It would be of the greatest interest to examine what effect lowering the false negative rate after SLNB only, could have on prognosis in these patients . We now know what the authors believe (Discussion line 257) but scientific evidence would be more interesting.

Reply: We thank the reviewer for the comment.

The limitations paragraph and conclusion were modified

The English language was professionally proofread and the whole manuscript was edited. Please find attached the proofread certification.

Reviewer 2 Report

My congratulations o the authors of this study. They focused on oncological outcomes instead of considering SNB as a diagnostic technique with its FNR, accuracy, etc.

The paper needs some minor changes before being accepted for publication.

Introduction

From my point of view, the introduction is a little confusing. I suggest reviewing the English edition.

Material and Methods:

According to methods, ALND was performed only if the intraoperative frozen section was positive. In that case, please explain how do you proceed in positive lymph nodes on H&E or when no drainage was observed.

Results

Table 1. According to table 1, 83% of patients with cN+ had postoperative radiotherapy. Current guidelines recommend radiotherapy in all patients with positive lymph nodes before or after NAT. Please, explain.

Table 2. According to table 2 median of the number SLN is the same in cN0 and cN+ and the p-value is 0.0004. I’m not an expert on statistics and I can’t understand why if the median is the same, the p-value is too low. Please, explain.

Table 3. As no differences are observed this table does not provide relevant information I suggest removing this table from the paper and offering it as supplementary material.

Discussion:

The first paragraph is too conclusive. This study shows that axillary relapses are less frequent than we expected from the results of SLNB accuracy in the NAQ setting.

In the same way, the data of this study are not enough strong. 160 patients are not enough to conclude “Sentinel lymph node biopsy is an acceptable procedure that can be safely performed in both cN0 and cN+ BC patients undergoing NAC who subsequently remained or became ycN0”

Author Response

POINT-BY-POINT REPLY TO REVIEWERS’ COMMENTS

We thank the editors and reviewers of CANCERS for the opportunity to reply to reviewers’ comments. We do believe that now the manuscript is much more precise and will be of interest to the readers of CANCERS. Please find our replies in this document in bold black.

Reviewers' Comments & Replies:

Reviewer #2: My congratulations o the authors of this study. They focused on oncological outcomes instead of considering SNB as a diagnostic technique with its FNR, accuracy, etc.

The paper needs some minor changes before being accepted for publication.

 Introduction

From my point of view, the introduction is a little confusing. I suggest reviewing the English edition.

Reply: We thank the reviewer for the comment.

The English language was professionally proofread and the whole manuscript was edited. Please find attached the proofread certification.

Material and Methods:

According to methods, ALND was performed only if the intraoperative frozen section was positive. In that case, please explain how do you proceed in positive lymph nodes on H&E or when no drainage was observed.

 Reply: We thank the reviewer for the comment.

No patient with an intra-operative negative SLN became positive at definitive histopathology examination.

This was added for clarification in the results section. Moreover, no case of no drainage was observed, at least 1 SLN was detected in all cases.

Results

Table 1. According to table 1, 83% of patients with cN+ had postoperative radiotherapy. Current guidelines recommend radiotherapy in all patients with positive lymph nodes before or after NAT. Please, explain.

Reply: We thank the reviewer for the comment.

In the cN+ population, adjuvant radiotherapy was recommended in all patients who underwent breast-conserving surgery. Additionally, in all patients who underwent mastectomy with previous T3-4 disease (chest wall adjuvant radiotherapy). Moreover, loco-regional/axillary radiotherapy was performed in all patients who underwent complete axillary lymph node dissection with ≥ 3 positive axillary lymph nodes.

Table 2. According to table 2 median of the number SLN is the same in cN0 and cN+ and the p-value is 0.0004. I’m not an expert on statistics and I can’t understand why if the median is the same, the p-value is too low. Please, explain.

 Reply: We thank the reviewer for the comment.

The median value expresses that in the majority of patients in both groups just 1 SLN was removed. In the range value it is expressed the min and max number of removed SLNs. Evidently, even if the median is the same between groups (1 SLN), there is a bigger sub-group in the cN+ population in which between 2 and 6 SLNs were removed compared with the cN0 population.

Table 3. As no differences are observed this table does not provide relevant information I suggest removing this table from the paper and offering it as supplementary material.

Reply: We thank the reviewer for the comment.

Table 3 was removed.

Discussion:

The first paragraph is too conclusive. This study shows that axillary relapses are less frequent than we expected from the results of SLNB accuracy in the NAQ setting.

In the same way, the data of this study are not enough strong. 160 patients are not enough to conclude “Sentinel lymph node biopsy is an acceptable procedure that can be safely performed in both cN0 and cN+ BC patients undergoing NAC who subsequently remained or became ycN0”

Reply: We thank the reviewer for the comment.

The discussion and the conclusion were modified.

Reviewer 3 Report

Dear authors,

I found your manuscript a well-organized piece of work. I thank you for the opportunity to review your
work. I do, however, suggest some minor cosmetics to be done. Please see below.

Comments:

Lines 2, 3, 4, 5 – slight change in title

Tite: Sentinel Lymph Node Biopsy in Breast Cancer Patients Undergoing Neo-adjuvant Chemotherapy: Clinical Experience with Node-negative and Node-positive Disease Prior to Systemic Therapy

Lines 17, 18 – small change, namely

The possibility of de-escalating axillary surgery in breast cancer patients undergoing neo-adjuvant chemotherapy is controversial.

Lines 23, 24, 25, 26 – small change, namely

This retrospective study is aimed to assess the reliability and safety of this approach by comparing the characteristics and oncological outcomes (e.g. overall survival) of cN0 and cN+ patients before neo-adjuvant chemotherapy and axillary surgery type.

Lines 29, 30, 31, 32 -small change, namely

SLNB is accepted in clinically node-negative (cN0) patients; however,its role in clinically node-positive (cN+) patients is debatable. Methods: A retrospective analysis of BC patients undergoing SLNB and NAC was done to evaluate the clinical significance of SLNB prior to NAC. This is accomplished by comparing the characteristics and oncological outcomes between cN0 and cN+ patients prior to NAC and type of axillary surgery.

Line 41,-small change, namely

This technique is not associated with worse oncological outcomes, thus de-escalation of axillary surgery can be proposed in the context of NAC.

Line 73, 74, 75, 76 -small change, namely

The aim of this retrospective study was to assess the reliability and clinical significance of lymphatic mapping with SLNB in the setting of NAC comparing the characteristics and oncological outcomes between cN0 and cN+ patients prior to chemotherapy and type of axillary surgery.

Line 79, 80, 81 -small change, namely

A retrospective review of all the consecutive BC patients scheduled for NAC and SLNB at the Breast Unit of IRCCS Humanitas Research Hospital (Milan, Italy), between November 2008 and December 2021 was done. Each patient gave the informed consent for surgery and clinical data collection.

Line 88,89 -small change, namely

...the management of every BC patient scheduled for NAC and SLNB.

Line 90 - small change, namely

...BC biological subtype and hospital protocols which are based on international guidelines.

Line 99, 100

·        Remove line concerning consent. It has been re-located. See above

Line 102, 103, 104, 105 – slight rewording

The protocol for lymphatic mapping was performed with the use of radioisotope alone, TAD was never used. For palpable tumors, the standard technique included a peri-tumoral injection of 99Technetium labeled radiocolloid, in efforts to replicate the intra-mammary lymphatic pathways potentially traversed by metastases.

Line 124 – slight rewording

Patients were selected from the institutional database (last ....

Line 141 – slight rewording

Statistical significance was set at p < 0.05 and all statistical tests were two-tailed.

·        Remove reference to IBM software.

Line 150 -rearrange-

cT2 150 tumors (cN0 67.9% versus cN+ 68.8%, p = 0.627) were present for the majority of patients.

Line 153 split sentence

...cN+ groups. However, cN+ patients were more frequently treated with ALND compared with cN0 patients (p = 0.001).

Line 173 – present tense

Axillary lymph nodes characteristics of patients are detailed in Table 2.

Line 192

Introduce the table here and remove lines 192 to 204—it is repetitive it is in the table. It becomes,namely;

Patients were divided according to their axillary status prior to NAC (cN0 versus cN+) and type of axillary surgery performed (SLNB-only versus ALND). Their oncological outcomes are summarized in Table 3. There was no statistically significant difference in terms of DFS, DDFS, OS, and BCSS between cN0 and cN+ patients prior to NAC (p = 0.180, p = 0.280, p = 0.181, p = 0.102, respectively), nor between patients treated with SLNB-only or ALND (p = 0.669, p = 0.665, p = 0.429, p = 0.776, respectively). Figure 1 and Figure 2 show the Kaplan-Meier recurrence and survival curves.

Figures – font too small (title, axis labels)

Figure captions: switch the alphabetical order to front, and correct spelling of breast in (d) namely

Line 216, 217, 218

Figure 1. (a) Disease-free survival, (b) distant disease-free survival, (c) overall survival, and (d) breast cancer-specific survival curves of breast cancer patients according to their axillary status prior to neo- adjuvant chemotherapy (cN0 versus cN+).

Line 220, 221, 222, 223

Figure 2. (a) Disease-free survival, (b) distant disease-free survival, (c) overall survival, and (d) breast cancer-specific survival curves of breast cancer patients undergoing neo-adjuvant chemotherapy according to the type of axillary surgery performed (sentinel lymph node biopsy-only versus axillary lymph node dissection).

Line 225 -rewording

The findings of the retrospective study contribute to the existing literature showing that lymphatic mapping with SLNB in the setting of NAC is oncologically safe for BC patients with cN0-cN+ disease. Axillary recurrence rates are consistent with those of previous retrospective analyses, ranging from 0 to 2.3% [26–29].

Line 250, 260, 261 – slight rewording.

Here, on the contrary, the oncological outcomes were reviewed and, in light of the results, can safely state that lymphatic mapping with SLNB can be performed in cN0-cN+ patients who are eligible for NAC. This concept has already been described by Kahler-Riberiro-Fontana et al. [26], who analyzed consecutive cT1-3, cN0-cN+ patients operated at the European Institute of Oncology who became of remained cN0 after NAC and underwent SLNB.

Line 275 – slight rewording

Second, the study had a short median follow-up.

Line 277,278, slight rewording

In conclusion, the breast unit is moving towards a de-escalation of axillary surgery even in the setting of NAC. The results demonstrated that lymphatic mapping with SLNB maintained its expected staging and prognostic role.

Author Response

POINT-BY-POINT REPLY TO REVIEWERS’ COMMENTS

We thank the editors and reviewers of CANCERS for the opportunity to reply to reviewers’ comments. We do believe that now the manuscript is much more precise and will be of interest to the readers of CANCERS. Please find our replies in this document in bold black.

Reviewers' Comments & Replies:

Reviewer #3: Dear authors,

I found your manuscript a well-organized piece of work. I thank you for the opportunity to review your
work. I do, however, suggest some minor cosmetics to be done. Please see below.

Comments:

Lines 2, 3, 4, 5 – slight change in title

Tite: Sentinel Lymph Node Biopsy in Breast Cancer Patients Undergoing Neo-adjuvant Chemotherapy: Clinical Experience with Node-negative and Node-positive Disease Prior to Systemic Therapy

Reply: We thank the reviewer for the comment.

The title was modified accordingly.

Lines 17, 18 – small change, namely

The possibility of de-escalating axillary surgery in breast cancer patients undergoing neo-adjuvant chemotherapy is controversial.

Reply: We thank the reviewer for the comment.

The text was modified accordingly.

Lines 23, 24, 25, 26 – small change, namely

This retrospective study is aimed to assess the reliability and safety of this approach by comparing the characteristics and oncological outcomes (e.g. overall survival) of cN0 and cN+ patients before neo-adjuvant chemotherapy and axillary surgery type.

Reply: We thank the reviewer for the comment.

The text was modified accordingly.

Lines 29, 30, 31, 32 -small change, namely

SLNB is accepted in clinically node-negative (cN0) patients; however,its role in clinically node-positive (cN+) patients is debatable. Methods: A retrospective analysis of BC patients undergoing SLNB and NAC was done to evaluate the clinical significance of SLNB prior to NAC. This is accomplished by comparing the characteristics and oncological outcomes between cN0 and cN+ patients prior to NAC and type of axillary surgery.

Reply: We thank the reviewer for the comment.

The text was modified accordingly.

Line 41,-small change, namely

This technique is not associated with worse oncological outcomes, thus de-escalation of axillary surgery can be proposed in the context of NAC.

Reply: We thank the reviewer for the comment.

The text was modified accordingly.

Line 73, 74, 75, 76 -small change, namely

The aim of this retrospective study was to assess the reliability and clinical significance of lymphatic mapping with SLNB in the setting of NAC comparing the characteristics and oncological outcomes between cN0 and cN+ patients prior to chemotherapy and type of axillary surgery.

Reply: We thank the reviewer for the comment.

The text was modified accordingly.

Line 79, 80, 81 -small change, namely

A retrospective review of all the consecutive BC patients scheduled for NAC and SLNB at the Breast Unit of IRCCS Humanitas Research Hospital (Milan, Italy), between November 2008 and December 2021 was done. Each patient gave the informed consent for surgery and clinical data collection.

Reply: We thank the reviewer for the comment.

The text was modified accordingly.

Line 88,89 -small change, namely

...the management of every BC patient scheduled for NAC and SLNB.

Reply: We thank the reviewer for the comment.

The text was modified accordingly.

Line 90 - small change, namely

...BC biological subtype and hospital protocols which are based on international guidelines.

Reply: We thank the reviewer for the comment.

The text was modified accordingly.

Line 99, 100

  • Remove line concerning consent. It has been re-located. See above

Reply: We thank the reviewer for the comment.

The text was modified accordingly.

Line 102, 103, 104, 105 – slight rewording

The protocol for lymphatic mapping was performed with the use of radioisotope alone, TAD was never used. For palpable tumors, the standard technique included a peri-tumoral injection of 99Technetium labeled radiocolloid, in efforts to replicate the intra-mammary lymphatic pathways potentially traversed by metastases.

Reply: We thank the reviewer for the comment.

The text was modified accordingly.

Line 124 – slight rewording

Patients were selected from the institutional database (last ....

Reply: We thank the reviewer for the comment.

The text was modified accordingly.

Line 141 – slight rewording

Statistical significance was set at p < 0.05 and all statistical tests were two-tailed.

  • Remove reference to IBM software.

Reply: We thank the reviewer for the comment.

The text was modified accordingly.

Line 150 -rearrange-

cT2 150 tumors (cN0 67.9% versus cN+ 68.8%, p = 0.627) were present for the majority of patients.

Reply: We thank the reviewer for the comment.

The text was modified accordingly.

Line 153 split sentence

...cN+ groups. However, cN+ patients were more frequently treated with ALND compared with cN0 patients (p = 0.001).

Reply: We thank the reviewer for the comment.

The text was modified accordingly.

Line 173 – present tense

Axillary lymph nodes characteristics of patients are detailed in Table 2.

Reply: We thank the reviewer for the comment.

The text was modified accordingly.

Line 192

Introduce the table here and remove lines 192 to 204—it is repetitive it is in the table. It becomes,namely;

Patients were divided according to their axillary status prior to NAC (cN0 versus cN+) and type of axillary surgery performed (SLNB-only versus ALND). Their oncological outcomes are summarized in Table 3. There was no statistically significant difference in terms of DFS, DDFS, OS, and BCSS between cN0 and cN+ patients prior to NAC (p = 0.180, p = 0.280, p = 0.181, p = 0.102, respectively), nor between patients treated with SLNB-only or ALND (p = 0.669, p = 0.665, p = 0.429, p = 0.776, respectively). Figure 1 and Figure 2 show the Kaplan-Meier recurrence and survival curves.

Reply: We thank the reviewer for the comment.

Lines 192 to 204 could not be removed because Reviewer 2 asked to remove Table 3.

Figures – font too small (title, axis labels)

Figure captions: switch the alphabetical order to front, and correct spelling of breast in (d) namely

Line 216, 217, 218

Figure 1. (a) Disease-free survival, (b) distant disease-free survival, (c) overall survival, and (d) breast cancer-specific survival curves of breast cancer patients according to their axillary status prior to neo- adjuvant chemotherapy (cN0 versus cN+).

Line 220, 221, 222, 223

Figure 2. (a) Disease-free survival, (b) distant disease-free survival, (c) overall survival, and (d) breast cancer-specific survival curves of breast cancer patients undergoing neo-adjuvant chemotherapy according to the type of axillary surgery performed (sentinel lymph node biopsy-only versus axillary lymph node dissection).

Reply: We thank the reviewer for the comment.

Figures and text were modified accordingly.

Line 225 -rewording

The findings of the retrospective study contribute to the existing literature showing that lymphatic mapping with SLNB in the setting of NAC is oncologically safe for BC patients with cN0-cN+ disease. Axillary recurrence rates are consistent with those of previous retrospective analyses, ranging from 0 to 2.3% [26–29].

Reply: We thank the reviewer for the comment.

The text was modified accordingly.

Line 250, 260, 261 – slight rewording.

Here, on the contrary, the oncological outcomes were reviewed and, in light of the results, can safely state that lymphatic mapping with SLNB can be performed in cN0-cN+ patients who are eligible for NAC. This concept has already been described by Kahler-Riberiro-Fontana et al. [26], who analyzed consecutive cT1-3, cN0-cN+ patients operated at the European Institute of Oncology who became of remained cN0 after NAC and underwent SLNB.

Reply: We thank the reviewer for the comment.

The text was modified accordingly.

Line 275 – slight rewording

Second, the study had a short median follow-up.

Reply: We thank the reviewer for the comment.

The text was modified accordingly.

Line 277,278, slight rewording

In conclusion, the breast unit is moving towards a de-escalation of axillary surgery even in the setting of NAC. The results demonstrated that lymphatic mapping with SLNB maintained its expected staging and prognostic role.

Reply: We thank the reviewer for the comment.

The text was modified accordingly.

Round 2

Reviewer 1 Report

In my opinion this retrospective study with very few events and not really comparable study groups is not good enough for backing the conclusions drawn by the authors.